# Dispersive FSO Performance Estimation with Gaussian Pulses and Laplace Modeled Time Jitter

P. J. Gripeos [1], D. Oreinos [1], D. Kriempardis [1], A. D. Tsigopoulos [2,*], E. Kapotis [1], A. Katsis [3] and H. E. Nistazakis [1]

[1] Section of Electronic Physics and Systems, Department of Physics, National and Kapodistrian University of Athens, 15784 Athens, Greece
[2] Department of Battle Systems, Naval Operations, Sea Studies, Navigation, Electronics and Telecommunications, Hellenic Naval Academy, Hadjikyriakou Ave, 18539 Piraeus, Greece
[3] Department of Social and Educational Policy, University of the Peloponnese, 20100 Korinthos, Greece
* Correspondence: atsigo@hna.gr; Tel.: +30-210-4581-606

**Abstract:** FSO communications tend to be one of most convenient, wireless, high-data-rate communications technologies of global telecom networking, and they are implemented and operated with low-cost resources. Despite their advantages, FSO systems' performance is delimited by several physical phenomena, which act on propagating signal beams through the atmospheric path. Among other effects, chromatic dispersion and time jitter affect the shape and the detection instant of the incoming optical pulse, respectively. This results in signal fading and probable misdetections, and the signal fades along the propagation path due to power losses. Particularly, chromatic dispersion affects the width of the longitudinal information pulse, while the stochastic nature of the time jitter effect is treated with the use of a statistical model for the instantly received irradiance of the detecting pulse at the corresponding time slot. In this study, the symmetrical Laplace distribution was chosen for weak time jitter effect emulation because of its symmetry in pulse detection before or after the center of the specific timeslot. Thus, the joint influence of all three effects could considered, including all the parameters involved. Moreover, new-closed-form mathematical expressions were derived in order to accurately estimate the availability and the reliability of the FSO links under consideration. Next, using the derived mathematical forms, performance outcomes were presented for typical parameter values for realistic FSO links.

**Keywords:** FSO links; Laplace-modeled time jitter; chromatic dispersion; chirped gaussian pulses; power losses; expected irradiance; outage probability; average bit error rate

## 1. Introduction

As one of the most favorable wireless technologies present in modern telecom networks globally, free space optical (FSO) communications are harmless for human health, riskless for signal integrity, unlicensed for spectrum usage, uncomplicated and inexpensive for equipment purchase, and high-data-rate transmissions since the installation of such systems requires low budget planning and the operation at infrared (IR) wavelengths allows large bandwidth exploitation [1–10]. However, the atmospheric channel, which the information carriers (i.e., the optical pulses) are propagated through, carries a lot of degrading effects that distort the pulse originally transmitted, thus diminishing the potentially promising efficiency of FSO systems [11–14].

Each effect acts separately and accumulatively along the propagation path. First of all, power losses are unavoidably involved as the electromagnetic wave propagates towards the receiver. By adopting a suitable loss coefficient, power loss fading can be easily determined [15,16]. Additionally, taking the pulse's non-monochromaticity into account, chromatic dispersion is generated due to the fact that each spectral component travels with a different velocity, resulting in pulse dimension distortion along the path [17–19].

Generally, for any given distance, the irradiance of the optical pulse is maximized at its center and decays symmetrically before and after that time point. The third effect studied in this work is related to time fluctuations of the incoming pulse's center around the detection point, which is known as the time jitter effect (TJ) [20–24]. Thus, the detector's output signal is always fluctuating as the incoming pulse is randomly time jittered (due to many factors), which depends on the propagation path, the electronic accuracy of the transmitter or the receiver, etc. This is especially problematic in very-high-data-rate FSO links. In this study, the TJ is modeled by the Laplace distribution [25–27], which describes early or delayed pulse arrivals indiscriminately and symmetrically.

An atmospheric effect which significantly impacts the performance of FSO links is atmospheric turbulence [1–10]. However, its influence has been studied broadly in the scientific literature and thus, it is out of the scope of the present work since we concentrate here only on the abovementioned effects which are dominant in very-high-data-rate communication systems with dispersive pulse propagation. However, the joint effect of all these factors could be studied further.

The remainder of this work is structured as follows: Section 2 contains the mathematical analysis of the physical effects, Section 3 refers to the availability and reliability of the link by means of the outage probability (OP) and average bit error rate (ABER), respectively. The corresponding numerical results are illustrated in Section 4, and the final conclusions are provided in Section 5.

## 2. System Model

Here, the link was implemented via optical, longitudinal, Gaussian modelled pulses, which were supposed to act as information carriers and mimic real FSO pulses, propagating and fading in either link direction. Furthermore, provided these pulses are transmitted in a memoryless, stationary, ergodic, dispersive, lossy, and noisy channel with independent and identically distributed (i.i.d.) intensity fading statistics suffered from additive white Gaussian noise (AWGN), $n$, characterized by zero mean value and variance equal to $N_0/2$, with $N_0$ being the noise power density, [28,29], then the electrical signal, $r$, at the receiver's output is given by the following equation [30–32]:

$$r = xs + n = x\eta I + n, \tag{1}$$

where $x$ modulates the "0" or "1" bit signal, $s = \eta I$ denotes the instantaneous beam intensity, $\eta$ represents the receiver's effective photo-current conversion ratio, and $I = I_r/I_0 = LBI_T$ is the instantaneous normalized irradiance if $I_0$ denotes the initially transmitted irradiance and $I_r$ denotes the finally received irradiance. The normalized irradiance can be assumed as a multifactorial product of the total power losses, $L$; the chromatic dispersion influence, $B$; and the TJ effect, $I_T$.

### 2.1. Total Power Losses Estimation

The total losses, $L$, can be elegantly described as a negatively exponential function of the distance propagated, $z$, [15,16,33,34]:

$$L(z) = \exp(-lz), \tag{2}$$

where $l$ denotes the loss coefficient expressed in power losses per distance unit. The values of the $l$ coefficient are related to the current weather conditions.

### 2.2. Chromatic Dispersion Influence Estimation

As the pulsewidth of the Gaussian pulse, $2T_{pw}$, of a propagating pulse along a dispersive channel spreads, its amplitude, $B$, is correspondingly reduced (and vice versa) in order for the energy under the pulse envelope to be maintained at any point of the path [18,24,35,36]. If $T$ symbolizes the time instant within a moving frame of a propagating pulse, $T = 0$ always corresponds to the pulse frame center. Equation (3) provides the

instantaneous, normalized irradiance, $I_t$, affected by chromatic dispersion and the TJ at the receiver [18,37,38]:

$$I_t(z, T) = B(z)I_T(z, T) = B(z)\exp\left\{-\left[T/T_{pw}(z)\right]^2\right\}, \tag{3}$$

where $B(z) = \left[1 + 2T_0^{-2}C\beta_2 z + T_0^{-4}\left(C^2 + 1\right)\beta_2^2 z^2\right]^{-1/2}$ and $T_{pw}(z) = \left[T_0^2 + 2C\beta_2 z + T_0^{-2}\right.$ $\left.\left(C^2 + 1\right)\beta_2^2 z^2\right]^{1/2}$, with $T_0$ being the original ($z = 0$) half-pulsewidth, $C$ representing the chirp parameter, $\beta_2\left[ps^2/km\right] = 1.75n(\omega)P_h\left(\pi c^2\lambda T_h\right)^{-1} \times 10^{15}$ representing the chromatic dispersion parameter, $c$ denoting the vacuum light speed, $\omega\left[rad/s\right] = 2\pi \times 10^6 \upsilon\lambda^{-1}$ being the angular frequency of a light beam operating at wavelength $\lambda[\mu m]$ and travelling at speed $\upsilon[m/s]$ through the medium, with refractive index $n(\lambda) = 1 + 77.6 \times 10^{-6}\left(1 + 7.52 \times 10^{-3}\lambda^{-2}\right)P_h/T_h$, which depends on the prevailing pressure $P_h[mbar] = 2.23 \times 10^{-6}\left(44.41 - h \times 10^{-3}\right)^{5.256}$ and temperature $T_h[K] = 288.19 - 6.49 \times 10^{-3}h$ conditions at the altitude $h[m]$ above the ground surface [18,37,38]. The TJ factor, $I_T$, is indeed related to the TJ effect since its value deviates from unity exclusively when $T \neq 0$, indicating a time-jittered pulse detection.

The case of $C = 0$ corresponds to an unchirped pulse, $C > 0$ indicates an up-chirped pulse, while $C < 0$ describes a down-chirped Gaussian pulse, which initially sharpens and further flattens. This changeover is placed at the critical distance, $z_{c1}$, [11,37]:

$$z_{c1} = \frac{|C|T_0^2}{\beta_2(C^2 + 1)}, \tag{4}$$

while the restoration of the original pulse dimensions ($I_{t,z=0,T=0} = I_{t,z_{c2},0}$) is placed at a second critical distance that is twice as much as the first one, $z_{c2} = 2z_{c1}$, [37].

### 2.3. TJ Influence Estimation

To estimation the influence of the TJ effect on each receiving pulse, the Laplace distribution model was used due to its symmetrical behavior that corresponds to the same probability to detect the information carrier, i.e., the longitudinal Gaussian pulse, before or after the correct moment. Its probability density function (pdf) with respect to $T$ is given as [11,25–27]:

$$f_T(T) = \frac{1}{2\sigma_T}\exp\left(-\frac{|T - \mu_T|}{\sigma_T}\right), \tag{5}$$

where $\sigma_T^2$ indicates the distribution variance, which models the strength of the TJ, and $\mu_T$ denotes its mean value, which can be equal to zero since the TJ effect is modelled here by symmetrical distribution.

Combining all the studied effects, their joint instantaneous irradiance, $I$, is given as:

$$I(z, T) = L(z)B(z)I_T(z, T) = LB\exp\left[-\left(T/T_{pw}\right)^2\right]. \tag{6}$$

Thus, it turns out that $T = \pm T_{pw}\sqrt{\ln(LB/I)}$, which means (5) can be rewritten as:

$$f_I(I) = \frac{f_T\left[\sqrt{\ln(LB/I)}\right]}{|dI/dT|_{T=+T_{pw}\sqrt{\ln(LB/I)}}} + \frac{f_T\left[-\sqrt{\ln(LB/I)}\right]}{|dI/dT|_{T=-T_{pw}\sqrt{\ln(LB/I)}}}, \tag{7}$$

Resulting in the following expression:

$$f_I(I) = \frac{g_T\left\{\exp\left[\sqrt{\ln(LB/I)}\right]\right\}^{-g_T}}{2\sqrt{\ln(LB/I)}I}, \tag{8}$$

where $g_T = T_{pw}/\sigma_T$.

Next, the expected irradiance of $I$ is computed by [39]:

$$E[I] = \int_0^{LB} I f_I(I) dI. \tag{9}$$

By applying Equation (8) into Equation (9) and using an RV transformation of the form: $y_I = \sqrt{\ln(LB/I)}$, Equation (9) gives the following outcome:

$$E[I] = \sqrt{\pi} g_T LB \exp\left(g_T^2/4\right) erfc(g_T/2)/2, \tag{10}$$

where *erfc*(.) stands for the complementary error function [40].

## 3. FSO Link Performance

### 3.1. Link's Availability Performance Estimation

A very significant quantity for the availability performance of the link is the outage probability (OP), which expresses the probability that the signal-to-noise ratio (SNR) is below of a given threshold, $\gamma_{th}$, within a specified time interval [31]. Because the incoming signal is continually changing over time, the instantaneous electrical SNR is given as $\gamma = (\eta I)^2/N_0$, while the average SNR is estimated as $\overline{\gamma} = (\eta E[I])^2/N_0$. Additionally, the detector has a constrained period of time to detect the incoming signal (known as time slot), which lasts for $T_{sl}$. Thus, the minimum expected irradiance at the edges of the time slot is: $I_{\min} = LB \exp\left\{-\left[T_{sl}/(2T_{pw})\right]^2\right\}$. Lastly, the time slot duration is a function of the bit rate, $R$, which depends on the modulation scheme used. Assuming an OOK modulation scheme, these quantities are inversely proportional to each other $\left(T_{sl} = R^{-1}\right)$. Thus, the probability of outage is estimated as [31]:

$$P_{out} = F_\gamma(\gamma_{th}) = \int_{\gamma_{\min}}^{\gamma_{th}} f_\gamma(\gamma) d\gamma, \tag{11}$$

where $f_\gamma(\gamma) = f_I(\gamma) dI/d\gamma$, with $I = \sqrt{\gamma/\overline{\gamma}} E[I_{t,l}]$, $\gamma_{\min} = \overline{\gamma}(I_{\min}/E[I])^2$, and $\gamma_{th} = \overline{\gamma}(I_{th}/E[I])^2$.

Next, the OP versus the normalized average SNR, $\overline{\gamma}/\gamma_{th}$, is given from Equations (8) and (11) as:

$$P_{out}(\overline{\gamma}/\gamma_{th}) = \frac{g_T}{4} \int_{\gamma_{\min}}^{\gamma_{th}} \frac{\exp\left[-g_T\left|\sqrt{\ln\left(LB\sqrt{\overline{\gamma}/\gamma}/E[I]\right)}\right|\right]}{\gamma\sqrt{\ln\left(LB\sqrt{\overline{\gamma}/\gamma}/E[I]\right)}} d\gamma. \tag{12}$$

After mathematical manipulations and the use of a random variable (RV) transformation $y_\gamma = \sqrt{\ln\left(LB\sqrt{\overline{\gamma}/\gamma}/E[I]\right)}$, Equation (12) is solved in the following form:

$$P_{out}(\overline{\gamma}/\gamma_{th}) = \exp\left[-g_T\sqrt{\ln\left(\frac{LB}{E[I]}\sqrt{\frac{\overline{\gamma}}{\gamma_{th}}}\right)}\right] - \exp(-g_T h_T), \tag{13}$$

where $h_T = T_{sl}/(2T_{pw})$.

### 3.2. Link's Reliability Performance Estimation

The reliability of a link can be estimated by the BER, which expresses the percentage of the erroneous bit from a large bit sequence, received in a given time interval, or, equivalently,

the probability of a bit to be erroneously received. The instantaneous BER for an OOK modulation is given as, [41–43]:

$$P_e(I) = \frac{1}{2}erfc\left(\frac{\sqrt{2}\eta I}{4\sqrt{N_0}}\right) \Leftrightarrow P_e(\gamma) = \frac{1}{2}erfc\left(\frac{\sqrt{2\gamma}}{4}\right), \tag{14}$$

while the average BER (ABER) is given as [41–43]:

$$\overline{P}_e(I) = \int_{I_{\min}}^{BL} P_e(I)f_I(I)dI \Leftrightarrow \overline{P}_e\left(\overline{\gamma}_j\right) = \int_{\gamma_{\min}}^{\gamma_{\max}} P_e(\gamma)f_\gamma(\gamma)d\gamma, \tag{15}$$

where $\gamma_{\max} = \overline{\gamma}(BL/E[I])^2$.

Substituting (8) and (14) into (15), the ABER, as a function of average SNR, is given as:

$$\overline{P}_e(\overline{\gamma}) = \frac{g_T}{4}\int_{\gamma_{\min}}^{\gamma_{\max}} erfc\left(\frac{\sqrt{2\gamma}}{4}\right)\frac{\exp\left[-g_T\left|\sqrt{\ln\left(LB\sqrt{\overline{\gamma}/\gamma}/E[I]\right)}\right|\right]}{\gamma\sqrt{\ln\left(LB\sqrt{\overline{\gamma}/\gamma}/E[I]\right)}}d\gamma. \tag{16}$$

Given that $erfc(x) = 2Q\left(\sqrt{2}x\right)$, $Q(x) \approx \left[\exp(-x^2/2) + 3\exp(-2x^2/3)\right]/12$, [44], and once again the $y_\gamma$ RV transformation, (16) is rewritten in the following form:

$$\overline{P}_e(y_\gamma) = \frac{g_T}{12}\int_0^{h_T}\exp(-g_Ty_\gamma)\left\{\exp\left[-\frac{\overline{\gamma}(LB)^2}{8E^2[I]}\exp\left(-2y_\gamma^2\right)\right] + 3\exp\left[-\frac{\overline{\gamma}(LB)^2}{6E^2[I]}\exp\left(-2y_\gamma^2\right)\right]\right\}dy_\gamma. \tag{17}$$

Replacing the external exponentials by applying $\exp(x) = \sum_{k=0}^{+\infty} x^k/k!$, the ABER expression is finally obtained:

$$\overline{P}_e(\overline{\gamma}) = \frac{\sqrt{\pi/2}g_T}{24}\sum_{k=0}^{+\infty}\frac{\left[-\overline{\gamma}(LB/E[I])^2\right]^k}{k!\sqrt{k}}\left(\frac{1}{8^k} + \frac{3}{6^k}\right)\exp\left(\frac{g_T^2}{8k}\right)\left[erf\left(\frac{g_T + 4kh_T}{2\sqrt{2k}}\right) - erf\left(\frac{g_T}{2\sqrt{2k}}\right)\right]. \tag{18}$$

## 4. Numerical Results

In this section, the numerical outcomes from the model equations above are presented using indicative realistic parameter values. It should be mentioned here that using the above-derived mathematical expressions, the FSO link's performance can be accurately estimated for any parameter values set for each optical wireless communication system under consideration. Thus, we assumed a horizontal terrestrial FSO link operating at 0.85 µm with a transmission rate of 7 Gbps at a height of 30 m above the earth surface. The length of the link was assumed to be either 5 or 10 km, whereas two pulsewidths are examined, i.e., $T_{pw}$ = 25 ps or 15 ps. Additionally, the propagated pulses can be either up-chirped or down-chirped, i.e., $C = \pm15$, with power losses, i.e., $l = 0.1$ km$^{-1}$. Finally, the TJ influence was investigated for two strength parameter values, i.e., $\sigma_T$ = 30 ps and 50 ps.

The irradiance amplitude profiles are illustrated in Figure 1 for many pulses, corresponding to some interesting characteristic values of the studied model. At first, three pulse couples were distributed at three different TJ instants of 0 ps, 30 ps and 50 ps, representing cases without TJs and with TJs, which are equal to the two strength parameter values studied. To avoid misunderstandings, the time-jittered cases illustrated are neither fixed nor the most probable ones since the mean of the Laplace distribution has been set to zero. However, they are indicative cases of the respective TJ strength since the probability of a TJ to be between zero and either $\sigma_T$ is exactly $0.5[1 - \exp(-1)] \approx 31.6\%$. Hence, the green curves correspond to the original pulses at $z = 0$ for two different original pulsewidths,

while the black and the red curves represent the up-chirped and the down-chirped versions of the green pulses, respectively, received after 5 km (left panel) and 10 km (right panel) and affected by power losses action. Additionally, at the time instant of 71.4 ps, the time slot edge corresponding to the studied bit rate is illustrated. Remarkably, the height of each pulse case, as detected at the time slot center line, was crucial for both the availability and reliability of the respective case. Among the curves with the same original pulsewidth, the most time-jittered pulses had lower irradiance height at the time slot center, according to the distinguishing inset plots on the left of each panel. Additionally, among cases with the same TJ, the larger the original pulsewidth, the higher the irradiance at the time slot center. In all cases, the down-chirped pulses had a little less height than the corresponding up-chirped pulses at that point.

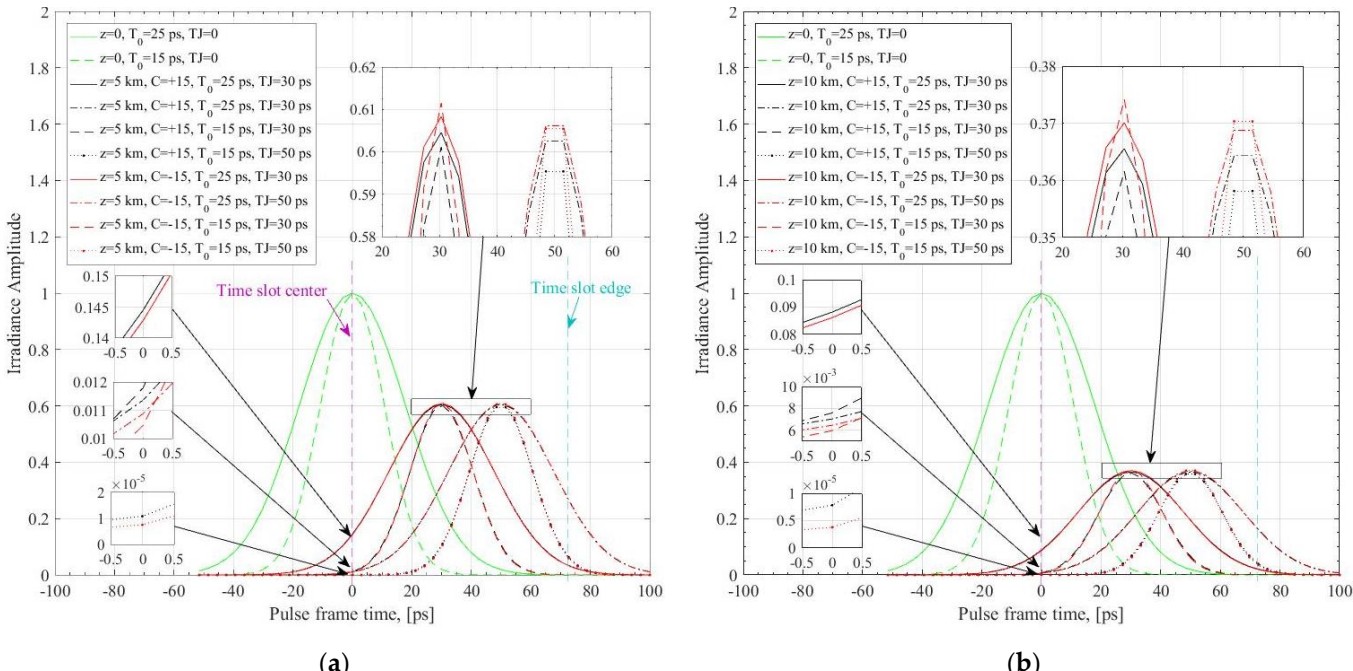

**Figure 1.** Irradiance amplitude profiles of some characteristic pulses within a time slot range with power losses ($l = 0.1$ km$^{-1}$) and various original pulsewidths, TJ strength parameters, and opposite chirp values for an FSO link of (**a**) $z = 5$ km and (**b**) $z = 10$ km. The time slot edge indicated at the time instant of 71.4 ps corresponds to 7 Gbps.

Next, Figure 2 represents the OP as a function of the normalized average SNR, $\overline{\gamma}/\gamma_{th}$, with various original pulsewidths, TJ strength parameters, and opposite chirp values for an FSO link of either 5 km or 10 km. In all cases, the curves' slopes constantly declined, implying lower OP for larger normalized average SNR values. Additionally, larger pulsewidth cases had lower OP performance compared with the smaller ones. Additionally, weaker TJ cases were more available than stronger ones. Lastly, the average BER as a function of the average SNR is depicted in Figure 3 for the same parameter combinations as in Figure 2. The curves here followed the monotony and arrangement of the corresponding curves of Figure 2. Hence, similar comments for the system's reliability can be made in accordance with the notes of the Figure 1 discussion.

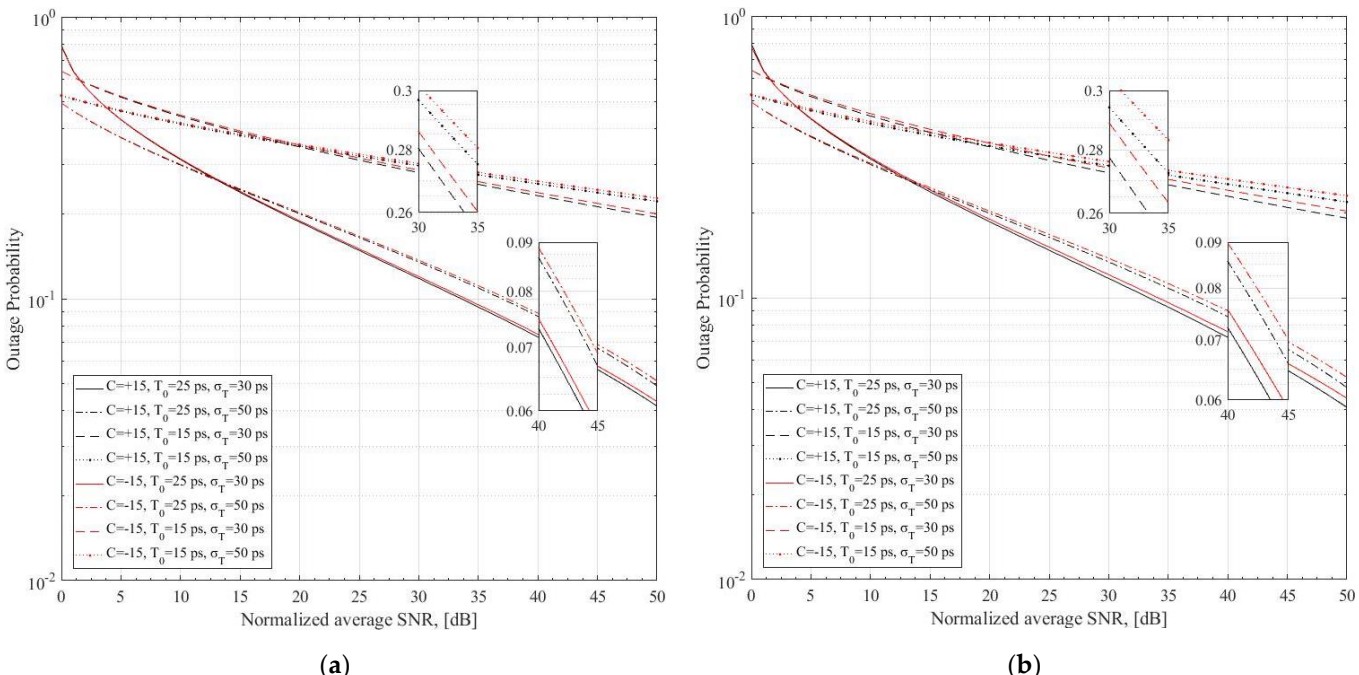

**Figure 2.** Outage probability as a function of the normalized average SNR with various original pulsewidths, TJ strength parameters, and opposite chirp values for an FSO link of (**a**) $z$ = 5 km and (**b**) $z$ = 10 km.

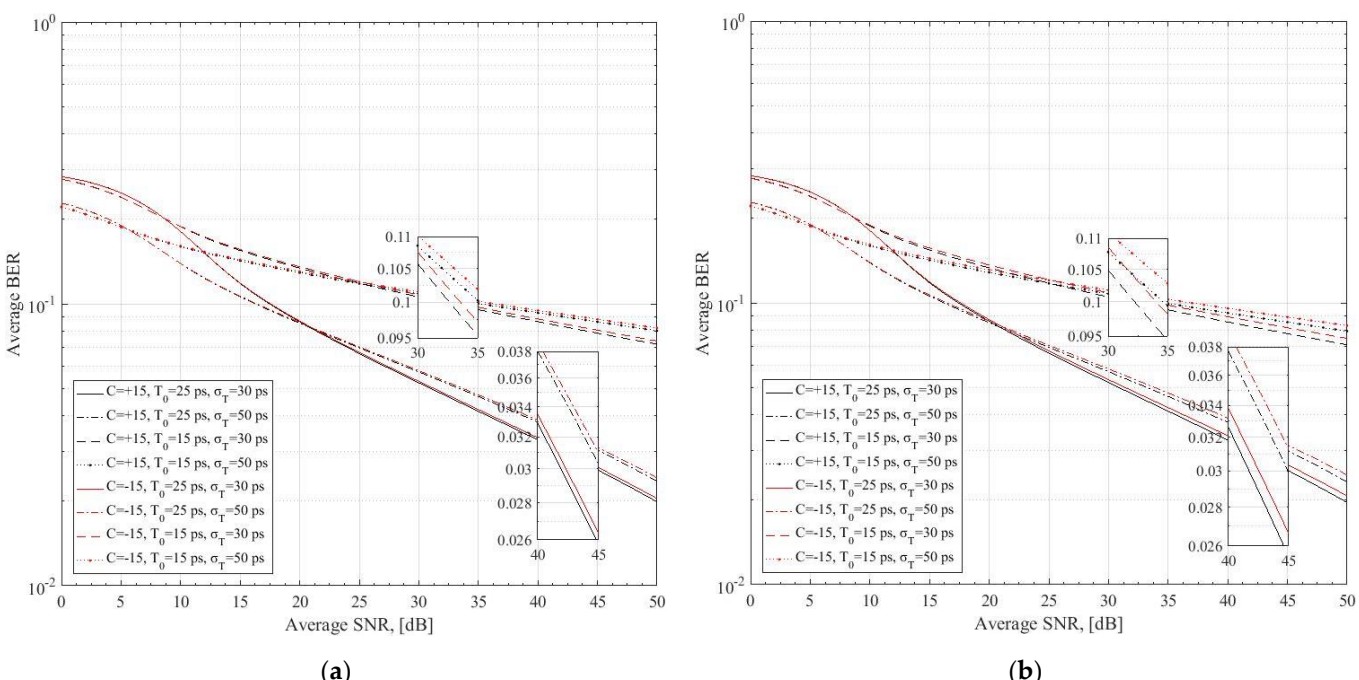

**Figure 3.** Average BER as a function of the average SNR with various original pulsewidths, TJ strength parameters, and opposite chirp values for an FSO link of (**a**) $z$ = 5 km and (**b**) $z$ = 10 km.

## 5. Conclusions

In this study, the availability and the reliability of FSO links was studied by assuming the propagation of longitudinal optical Gaussian pulses as information carriers in the dispersive atmosphere under the joint influence of chromatic dispersion and the Laplace-distributed TJ. Considering these effects, new closed-form mathematical expressions were derived for the accurate estimation of the outage probability and the average BER of each

specific FSO link. Using the derived mathematical expressions and typical parameter values, the corresponding indicative outcomes were plotted and the influence of each of the abovementioned effects was investigated. Furthermore, it should be mentioned that many other propagation effects, such as atmospheric turbulence, could be studied jointly to achieve more accurate outcomes.

**Author Contributions:** Conceptualization, P.J.G., D.O. and H.E.N.; methodology, P.J.G. and H.E.N.; software, P.J.G., E.K. and H.E.N.; validation, P.J.G., D.O., D.K., A.D.T. and H.E.N.; formal analysis, P.J.G., D.K. and H.E.N.; investigation, P.J.G., D.O., A.K. and H.E.N.; resources, P.J.G. and H.E.N.; writing—original draft preparation, P.J.G. and H.E.N.; writing—review and editing, P.J.G., A.D.T., E.K., A.K. and H.E.N.; supervision, P.J.G., A.D.T., A.K. and H.E.N.; project administration, A.D.T. and H.E.N. All authors have read and agreed to the published version of the manuscript.

**Funding:** This research received no external funding.

**Data Availability Statement:** Not applicable.

**Conflicts of Interest:** The authors declare no conflict of interest.

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
