# Peer review of "Dispersive FSO Performance Estimation with Gaussian Pulses and Laplace Modeled Time Jitter"

_computation, doi:10.3390/computation11010006_

Round 1
Reviewer 1 Report
This manuscript, entitled “Dispersive FSO Performance Estimation with Gaussian Pulses 2 and Laplace Modeled Time Jitter,” deduces the FSO transmission model, including three effects: power loss, chromatic dispersion, and timing jitter. Considering these effects is important to design practical FSO communication systems. However, I have some concerns about the manuscript, as listed below. So, I disagree with publishing the manuscript as a regular article of Computation.
(1) The proposed model does not contain atmospheric fading, which may be expressed by the log-normal and gamma-gamma distributions typically. I think the authors should contain this effect when considering terrestrial FSO communication as they do in Section 4.
(2) The figures are too small to examine carefully. The authors should enlarge the figures or increase their resolution.
(3) The statement from line 137 to line 139 seems nonsense. This should be removed.
(4) Equation (6) is referred to at line 129, which I guess wrong: Equation (5) is correct.
Author Response
Replies to the Comments of Editor and Reviewers
Manuscript ID: computation-2050206 Manuscript title: Dispersive FSO Performance Estimation with Gaussian Pulses and Laplace Modeled Time Jitter Authors:P.J. Gripeos, D. Oreinos, D. Kriempardis, A.D. Tsigopoulos, E. Kapotis, A. Katsis and H.E. Nistazakis
First of all, we would like to thank the Editor and the anonymous Reviewers for their valuable comments and their time and effort in assessing our manuscript. Please find our detailed answers to the comments, as well as indications of changes/additions made in the new version of the manuscript. In order to support the review process, the changes/additions have been highlighted in red color.
Reviewers’ Comments:
Reviewer #1:
- What are the differences and the advantages by Gaussian Pulses and Laplace Modeled Time Jitter and other pulses and other models?
AND
What are the significance by Gaussian Pulses and Laplace Modeled Time Jitter?
Authors: Thank you very much for the comments. In the revised version of our manuscript, we are trying to explain more detailed that the longitudinal Gaussian pulse propagation has been studied because this kind of pulses can better emulate the realistic pulses which are used in the FSO links. Of course, other kind of pulses can also be used for the emulation of the bit pulse propagation, e.g. supergaussian pulses etc, but the Gaussian ones can be assumed to be very close to the realistic ones.
On the other hand, the Laplace model for the Time Jitter effect has been used because it presents the insight of the jitter effect, i.e. there is no difference, in the most of the cases, to detect the pulse earlier or later, especially the very short pulses of the very high data rate systems, and the probability decays as the detection moment is getting away for the center of the pulse. It is clear that for the specific effect there are many others models which can be used to emulate the behavior of the time Jitter, e.g. normal distribution, etc, depending on each specific case under consideration.
- In the line 153 of page 4, “is given form” should be “is given from”.
Authors: Thank you for your comment. This mistake has been corrected in the new version of our work.

Reviewer 2 Report
In this manuscript (Dispersive FSO Performance Estimation with Gaussian Pulses and Laplace Modeled Time Jitter), the authors discuss FSO communications tend to be one of most modern, convenient, wireless, high data rate communications technologies of the global telecom networking, implemented and operated by low cost resources. Despite their potentially high throughput, in fact, FSO systems’ performance is delimited by several physical phenomena, which act on the propagating signal beams through the atmosphere. Among other effects, chromatic dispersion and time jitter affect the shape and the detection instant of the incoming, optical pulse, respectively, ensuing signal fading and probable misdetections, while the signal is exponentially fading along the propagation path, due to power losses. Particularly, the chromatic dispersion is regulated by the sign and the magnitude of the
pulse’s chirp parameter. Also, the stochastic nature of the time jitter effect is treated with the use of
a statistical model for the instantly receiving irradiance of the detecting pulse at the corresponding
time slot center. In this work, the symmetrical Laplace distribution is chosen for weak time jitter effect emulation. However, the joint influence of all three effects is taken into account, generating the joint probability density function, including all the parameters involved. Next, the availability and the reliability performance are thoroughly examined by means of outage probability and average bit error rate, respectively. The results produced for typical FSO links’ parameters, show that both performance quantities are depending on the time jitter strength, the chirp sign, the losses coefficient and the propagating distance to the receiver’s detector.
The topic is interesting and hot, and I can recommend the publication of the manuscript in Computation after major revisions:
1, What are the differences and the advantages by Gaussian Pulses and Laplace Modeled Time Jitter and other pulses and other models?
2, What are the significance by Gaussian Pulses and Laplace Modeled Time Jitter?
3. In the line 153 of page 4, “is given form” should be “is given from”.

Author Response
Replies to the Comments of Editor and Reviewers
Manuscript ID: computation-2050206 Manuscript title: Dispersive FSO Performance Estimation with Gaussian Pulses and Laplace Modeled Time Jitter Authors:P.J. Gripeos, D. Oreinos, D. Kriempardis, A.D. Tsigopoulos, E. Kapotis, A. Katsis and H.E. Nistazakis
First of all, we would like to thank the Editor and the anonymous Reviewers for their valuable comments and their time and effort in assessing our manuscript. Please find our detailed answers to the comments, as well as indications of changes/additions made in the new version of the manuscript. In order to support the review process, the changes/additions have been highlighted in red color.
Reviewers’ Comments:
Reviewer #2:
- Page 2: Line 70: “In this study, the TJ is modeled by the Laplace distribution [25]–[27]”: The Time Jitter effect statistics is assumed to follow the Laplace distribution: the authors should comment based on some simple physics of the propagation that why TJ statistics was assumed to follow the Laplace distribution. Just one or two sentences should be added so that the readers have the clear understanding of the basic start of the problem investigated in their research.
Authors: Thank you very much for the comments. In the revised version of our manuscript, we are trying to explain more detailed that the Laplace model has been used for the Time Jitter effect, because it presents the insight of the jitter effect, i.e. there is no difference, in the most of the cases, to detect the pulse earlier or later, especially the very short pulses of the very high data rate systems, and the probability decays as the detection moment is getting away for the center of the pulse. It is clear that for the specific effect there are many other models which can be used to emulate the behavior of the time Jitter, e.g. normal distribution model, etc, depending on each specific case under consideration.
- Page 5: Line 176: The optical propagation selected for this research is 30 m above the ground and for two path lengths, 5 and 10 km. For this type of path, atmospheric turbulence plays an important role. My question is whether the turbulence effects are lumped in the two different strength parameter’s values of 30 ps or 50 ps? If not, then at least mention the atmospheric effects. Also, the horizontal path chosen is probably assumed to be valid for homogeneous turbulent, which can be true for 5 km, but may not be strictly true for 10 km where some non-uniform turbulence can play some effect in this research.
Authors: Thank you very much for the comment. In the revised version of our manuscript, we tried to explain better that the atmospheric turbulence effect plays a very significant role at the performance of the optical pulse propagation in the cases of terrestrial FSO links. Especially in long link lengths, as the reviewer mention.
However, the very significant atmospheric turbulence effect in FSO links, has been already studied in many published works.
Thus, in this work we tried to concentrate our investigation on three other very significant effects which have not been studied extensively in many works, although they play very significant roles at the FSO links’ performance, especially the very high data rate ones. More specifically, we studied the influence of the chromatic dispersion effect, which is significant for very short longitudinal pulses, i.e. very high data rates FSO communication systems, the time jitter effect which affects strongly the performance of very fast links and the propagation losses, as well.
It is clear that the reviewer’s comment is very significant, and we think that the joint investigation of the influence of the atmospheric turbulence along with the above mentioned effects could be the next step.
- Page 7: Figures 2 and 3: For both the “Outage Probability” and “Average BER” to have the typical desired values between 10-1 and 10-2 (or less if possible with proper lasercom designs), the normalized average SNR [dB] of about 40 to 45 dB needed. The authors should mention a typical how much diode laser power operating at wavelength of 0.85 micrometer needed for this type of lasercom system to design.
AND
It would be helpful if the authors can mention about the maximum achievable data rate (in their example 7 Gbps) possible based on their results.
AND
Can their results described in this paper be extended to a typical transmitting wavelength of 1.55 micrometer?
Authors: Thank you very much for the comments. This work is mainly theoretical, and we are concentrating to extract new closed form mathematical expressions in order to estimate accurately the influence of the above-mentioned effects, i.e. group velocity dispersion, propagation losses and time jitter, at the FSO links performance. Thus, the obtained expressions can be used for any type of links with specific parameters values depending on the specific realistic FSO link under consideration. Thus, any parameter values can be used such as, operational wavelength, Time Jitter strength, etc and the FSO system’s performance can be estimated. In the numerical results of our work, we just present some practical and realistic cases to present the results that we obtain from the new derived equations, in order to present the accuracy of our outcomes. It is clear that, any other set of values can be used depending on each specific case under consideration.
- Are the results valid for two-way communications? Note that the optical path characteristics are little bit different for two different directions (from the atmospheric turbulence point of view).
Authors: Thank you very much for the comment. In the revised version of our manuscript, we explain more detailed that the influence of the three effects examined here, cause the same outcomes in both propagation directions.

Reviewer 3 Report
REVIEW of MDPI Paper-12-12-2022- Dispersive FSO…
Date: 12-12-2022
Comments:
Page 2: Line 70: “In this study, the TJ is modeled by the Laplace distribution [25]–[27]”: The Time Jitter effect statistics is assumed to follow the Laplace distribution: the authors should comment based on some simple physics of the propagation that why TJ statistics was assumed to follow the Laplace distribution. Just one or two sentences should be added so that the readers have the clear understanding of the basic start of the problem investigated in their research.
Page 5: Line 176: The optical propagation selected for this research is 30 m above the ground and for two path lengths, 5 and 10 km. For this type of path, atmospheric turbulence plays an important role. My question is whether the turbulence effects are lumped in the two different strength parameter’s values of 30 ps or 50 ps? If not, then at least mention the atmospheric effects. Also, the horizontal path chosen is probably assumed to be valid for homogeneous turbulent, which can be true for 5 km, but may not be strictly true for 10 km where some non-uniform turbulence can play some effect in this research.
Page 7: Figures 2 and 3: For both the “Outage Probability” and “Average BER” to have the typical desired values between 10-1 and 10-2 (or less if possible with proper lasercom designs),the normalized average SNR [dB] of about 40 to 45 dB needed. The authors should mention a typical how much diode laser power operating at wavelength of 0.85 micrometer needed for this type of lasercom system to design.
General Comments:
- It would be helpful if the authors can mention about the maximum achievable data rate (in their example 7 Gbps) possible based on their results.
- Are the results valid for two-way communications? Note that the optical path characteristics are little bit different for two different directions (from the atmospheric turbulence point of view).
- Can their results described in this paper be extended to a typical transmitting wavelength of 1.55 micrometer?
Author Response
Replies to the Comments of Editor and Reviewers
Manuscript ID: computation-2050206 Manuscript title: Dispersive FSO Performance Estimation with Gaussian Pulses and Laplace Modeled Time Jitter Authors:P.J. Gripeos, D. Oreinos, D. Kriempardis, A.D. Tsigopoulos, E. Kapotis, A. Katsis and H.E. Nistazakis
First of all, we would like to thank the Editor and the anonymous Reviewers for their valuable comments and their time and effort in assessing our manuscript. Please find our detailed answers to the comments, as well as indications of changes/additions made in the new version of the manuscript. In order to support the review process, the changes/additions have been highlighted in red color.
Reviewers’ Comments:
Reviewer #3:
- The proposed model does not contain atmospheric fading, which may be expressed by the log-normal and gamma-gamma distributions typically. I think the authors should contain this effect when considering terrestrial FSO communication as they do in Section 4.
Authors: Thank you very much for your comment. In the revised version of our manuscript, we tried to explain better that the atmospheric turbulence effect plays a very significant role at the performance of the optical pulse propagation in the cases of terrestrial FSO links, especially in long link lengths, as the reviewer mentioned.
However, the very significant atmospheric turbulence effect in FSO links, has been already studied in many published works.
Thus, in this work we tried to concentrate our investigation on three other very significant effects which have not been studied extensively in many works, although they play very significant roles at the FSO links’ performance, especially the very high data rate ones. More specifically, we studied the influence of the chromatic dispersion effect, which is significant for very short longitudinal pulses, i.e. FSO links with very high data rates, the time jitter effect which affects strongly the performance of very fast links and the propagation losses, as well.
It is clear that, the reviewer’s comment is very significant, and we think that the joint investigation of the influence of the atmospheric turbulence effect along with the above-mentioned effects could be the next step.
- The figures are too small to examine carefully. The authors should enlarge the figures or increase their resolution.
Authors: Thank you very much for your comment. With respect to the horizontal margin of the page, in the revised version of our manuscript, all figures have been enlarged by almost 22% in each dimension.
- The statement from line 137 to line 139 seems nonsense. This should be removed.
Authors: Thank you very much for your apposite comment. These lines have been removed in the new version of our work.
- Equation (6) is referred to at line 129, which I guess wrong: Equation (5) is correct.
Authors: Thank you very much once again. The new version also includes this replacement.

Round 2
Reviewer 1 Report
I appreciate the authors' great efforts to improve their manuscript based on my comments. I agree to publish the manuscript.
Reviewer 2 Report
The revised version of this manuscript can be accepted for publication.